# Evaluation of Different Models for Non-Destructive Detection of Tomato Pesticide Residues Based on Near-Infrared Spectroscopy

**DOI:** 10.3390/s21093032

**Published:** 2021-04-26

**Authors:** Araz Soltani Nazarloo, Vali Rasooli Sharabiani, Yousef Abbaspour Gilandeh, Ebrahim Taghinezhad, Mariusz Szymanek

**Affiliations:** 1Department of Biosystem Engineering, Faculty of Agriculture and Natural Resources, University of Mohaghegh Ardabili, Ardabil 56199-11367, Iran; arazsoltani@uma.ac.ir (A.S.N.); abbaspour@uma.ac.ir (Y.A.G.); 2Department of Agricultural Engineering and Technology, Moghan College of Agriculture and Natural Resources, University of Mohaghegh Ardabili, Ardabil 56199-11367, Iran; e.taghinezhad@uma.ac.ir; 3Department of Agricultural, Forest and Transport Machinery, University of Life Sciences in Lublin, Street Głęboka 28, 20-612 Lublin, Poland; mariusz.szymanek@up.lublin.pl

**Keywords:** pesticide residues, spectroscopy, PLS, soft computing, algorithm

## Abstract

In this study, the possibility of non-destructive detection of tomato pesticide residues was investigated using Vis/NIRS and prediction models such as PLSR and ANN. First, Vis/NIR spectral data from 180 samples of non-pesticide tomatoes (used as a control treatment) and samples impregnated with pesticide with a concentration of 2 L per 1000 L between 350–1100 nm were recorded by a spectroradiometer. Then, they were divided into two parts: Calibration data (70%) and prediction data (30%). Next, the prediction performance of PLSR and ANN models after processing was compared with 10 spectral preprocessing methods. Spectral data obtained from spectroscopy were used as input and pesticide values obtained by gas chromatography method were used as output data. Data dimension reduction methods (principal component analysis (PCA), Random frog (RF), and Successive prediction algorithm (SPA)) were used to select the number of main variables. According to the values obtained for root-mean-square error (RMSE) and correlation coefficient (R) of the calibration and prediction data, it was found that the combined model SPA-ANN has the best performance (RC = 0.988, RP = 0.982, RMSEC = 0.141, RMSEP = 0.166). The investigational consequences obtained can be a reference for the development of internal content of agricultural products, based on NIR spectroscopy.

## 1. Introduction

Tomato (*Solanum lycopersicum*) is one of the most widely used crops in the world, which is rich in antioxidants such as carotenoids, total phenols, vitamin E, and vitamin C [1]. Related empirical studies have shown that vitamin C affects the human immune system and prevents diseases such as Alzheimer’s [2]. In addition, the prevention of illnesses by fruits and vegetables also depends on antioxidants [3].

Tomatoes need intensive pest management due to their low resistance to pests and diseases. The need to use pesticides can leave harmful residues in the product. Organophosphorus pesticides can be stable for a considerable time even after washing and cooking in the product if used without observing its pre-harvest interval [4,5,6].

Today, many countries have restricted the use of pesticides, requiring the pesticide maximum residue limit (MRL) in food [7], and the amount is specified for each crop [8,9,10,11]. Currently, there are several methods for determining the concentration of pesticides, including GC, HPLC, thin layer chromatography, and capillary electrophoresis [12]. However, due to time constraints and high costs, it is not possible to use these methods to control all products [13].

Many studies are currently underway to develop safe, rapid, reliable, and low-cost methods for determining pesticide residues that can prevent the use of organic solvents and reduce operator exposure to toxic substances. Spectroscopy-based methods are a potential method that can solve the problems mentioned above.

NIRS is suitable non-destructive method for quantitative and qualitative analysis in agriculture, chemistry, medicine, and other sciences [14,15,16,17,18]. This technique is faster and cheaper than conventional methods and environmentally friendly and can usually be used without the need to prepare samples [19,20,21,22]. This technology is based on the absorption of radiation in the infrared region near the electromagnetic spectrum, which can be used to control the quality of food products [23,24,25]. Furthermore, in some studies, this technology has been used to detect pesticide residues in agricultural products [26,27,28].

Jun et al. [29] examined cadmium residue in tomato leaves using hyperspectral imaging. In this method, WT and LSSVR were used to choose the best wavelength and create a detection model. The best prediction performance for the detection of cadmium (Cd) content in tomato leaves was obtained using the second derivative preprocessing method.

Chen et al. [30] used NIRS to determine organophosphate chemicals. PLSR was used to create the prediction models. The best prediction result was obtained using PLSR with MSC and the first derivative as the preprocessing method.

Fen et al. [31] used NIRS and ANN for non-destructive detection of a common pesticide on the Longan surface. The results showed that the correct diagnosis ratio was 93%.

Jiang et al. [32] combined deep learning and machine vision to predict the pesticide. The consequences showed that when the training epoch is 10, the precision of the test set detection will be 90.09% and the average picture bandwidth detection precision will be 95.35%.

Wei et al. [33] offered a technique for removing residues of pesticide in apple juice. This technique can precisely identify and classify data about residues of pesticide in apples.

Soltani et al. [18] used NIRS technology with multivariate regression analysis to predict pesticide residues in tomato. The best prediction results were obtained using the PLS model based on the smoothing + moving average method (Rcv = 0.92, RMSECV = 4.25). 

Xue et al. [34] used the PSO algorithm to predict dichlorvos residue on the orange surface by Vis-NIR spectroscopy. The PSO-PLS model was able to predict the dichlorvos residue with a correlation coefficient of 0.8732. They have stated that the selection of wavelengths through a PSO algorithm increases the ability to predict when using the PLS model.

According to previous studies, the NIRS can be used to predict pesticide residues from other crops. To the best of our knowledge, there is no research to determine the organophosphorus pesticides and their prediction methods in tomatoes.

Therefore, in this paper, we use NIRS and chemometric methods to create a prediction model without destruction to detect the tomato pesticide residues. Spectral data obtained with a spectroradiometer and reference data obtained by a gas chromatography equipment were used as input and output of the models used in this study, respectively. PCA, SPA, and RF algorithms were utilized to select the variable as input for artificial neural network (ANN) and PLSR. First, all spectral data without dimension reduction and then spectral data obtained from variable selection algorithms were used to predict the amount of pesticide in tomatoes. Then 8 combined modes (PLS, ANN, PCA-ANN, RF-ANN, SPA-ANN, PLS-PCA, PLS-RF, and PLS-SPA) were developed for pesticides residues prediction. The use of several algorithms for variable selection to predict organophosphorus pesticide in tomatoes has not been evaluated in previous research. New progress can be made in improving food quality by this investigation.

## 2. Materials and Methods

### 2.1. Sample Preparation

180 samples of tomatoes (Queen) were randomly harvested from a greenhouse where almost all their produce was uniform in size and stored until 5 °C until use. Pest control in tomatoes was non-chemical from the beginning of planting to the harvest stage. In order to achieve different pesticide residual concentrations, the samples were infected with Profenofos 40% (EC 40%) (C11H15BrClO3PS) with a Pre-Harvest Interval (PHI) of 14 days. Therefore, the solution of Profenofos pesticide with a concentration of 2 per 1000 L of water was prepared and sprayed on the samples. The samples were divided into 6 categories: The first group (P0) was used without any spraying as control and non-pesticide samples; second group two hours (P-2H); third group two days (P-2D); the fourth group is the same as the third category, except that it was washed after spraying (P-2D-W); the fifth group for one week (P-1W) and the sixth group for two weeks (P-2W) were subjected to VIS/NIR spectroscopy after spraying with the prepared solution. All samples reached equilibrium temperature in the laboratory before completing the measurements. 

### 2.2. Vis/NIR Spectroscopy

Vis/NIR spectroscopy tests was performed using a PS-100 spectroradiometer (Apogee Instruments, INC., Logan, UT, USA) with CCD detector, 2048 pixels, 1 nm resolution and halogen-tungsten light source in the wavelength range of 350–1100 nm. Prior to spectroscopy, black and white (reference) spectra were first defined and stored. In this way, first by turning off the light source, the dark spectrum was taken, then in the light source mode, a standard Teflon disk with the ability to reflect above 97 in the range of 300 to 1700 nm was used to achieve the reference spectrum. For each tomato sample from 4 different points of each sample with 8 scans, within the spectral range of the equipment used, spectroscopy was performed with software Spectra-Wiz Spectrometer OS v5.33 (c) 2014 and the data were recorded after averaging. To find the spectral regions in the pesticide solution a quartz cell and two single-stranded fiber optics P400-2-VIS-NIR was used (Figure 1) [35]. Reference measurements were performed one day after spectroscopic analysis [18].

### 2.3. Reference Measurements

After Vis/NIR spectroscopy, all tomatoes were prepared frozen to measure profenofos by gas chromatographic reference method (Agilent 5977A Series GC/MSD—Santa Clara, CA 95051, USA). To determine the retention time of the peak of the diagram obtained for Profenofos pesticide, the Profenofos standard material (95%) prepared from Agricultural Exir Company was injected into the chromatograph. For this purpose, sample preparation was performed according to the British standard BS EN 15662 [36,37]. First, 10 g of the homogenized sample was poured into a 50 mL centrifuge falcon. Then 10 mL of ethyl acetate, 1.9 mL of distilled water and 5 g of nitrogen sulfate were added and stirred for 1 min. It was then centrifuged at 5000 rpm for 5 min and 6 mL of the extract formed on top of the falcon was transferred to another glass falcon. It was shaken for 1 min and centrifuged at 5000 rpm for 5 min. Then 4 mL of the upper extract of glass was poured into another falcon and 50 μL of ethyl acetate was added. After filtration, 1 μL of extract was injected into the equipment. The run conditions of the gas chromatography equipment are fully described in Table 1.

### 2.4. Remove the Outlier Data

The Monte Carlo cross-validation method was used to remove outliers. This method can simultaneously detect spectral outliers and reference data [38]. Initially, the data were randomly divided into two categories: Calibration set (70%) and prediction set (30%). Then, PLS models were got with full cross-validation. When the RMSECV is minimized, the best number of PC of the model is achieved. Next, the statistical characteristic parameters of each model and the cumulative value of the sum of squares of predicted residual errors of each sample were determined [39,40]. In this paper, outlier data (20 samples) have been deleted by the method mentioned and the amount of R of the model has been improved from 0.8113 to 0.8609 after their removal. Table 2 shows the reference values (mean, standard deviation, and range) for the profenofos content (mg kg^−1^) in the tomato samples used in this study. As can be seen, the values ranged from n.d (Not detected) to 42.9 mg/kg.

### 2.5. Variable Selection Method

#### 2.5.1. Random frog (RF) Algorithm

The RF algorithm is generally used in the set of meta-heuristic algorithms. This algorithm is a useful wavelength selection method that calculates the probability of selection for each variable [40]. In short, the random frog algorithm consists of three steps [41,42]: (1) The random initialization of a subset of variable *V_0_* containing the variables *Q*; (2) creating a subset of the variable *V ** including the variable *Q **; accepting *V ** as *V*_1_ with a certain probability and considering *V*_0_ = *V*_1_; the above procedure is repeated until the end of N and (3) calculating the probability of selecting each variable that can be used as a measure of the importance of the variable. The schematic of the algorithm is shown in Figure 2.

Figure 3 shows the appropriate wavelengths attained by the RF algorithm. In order to have the large part of impressible data in the main spectrum, the selection threshold was determined experimentally by 20% trial and error method and the wavelengths above this selection threshold were selected as the number of characteristic wavelengths. Therefore, 28 wavelengths above the dotted line were used as the final wavelengths to predict pesticide residues in tomatoes.

#### 2.5.2. SPA

SPA is a forward selection method that uses simple operations in a vector space to minimize the linearity of variables. The useful variable can be selected in spectral data analysis for multivariate calibration using this new method. This technique is widely used in optimizing specific spectral wavelengths that evaluate variable subsets based on RMSEC [43]. According to the change curve of RMSEC in relation to the number of wavelengths, it was determined that by selecting 14 characteristic wavelengths, the value of RMSE attained a lowest value of 0.141 (Figure 4). Thus, 14 effective wavelengths were applied as input to the prediction model. The selected characteristic wavelength distributions across the whole spectrum are shown in Figure 5. Wavelengths close to 650–700, 750–800 and 960–1000 were chosen to build the model. These wavelengths were in some cases like the wavelengths of the RF algorithm.

#### 2.5.3. PCA

PCA is one of the most widely used multivariate statistical methods in chemistry [44,45]. The corresponding mathematical model for PCA is based on the decomposition of matrix *X* into score matrix *n* × *A* (*T*) and loading matrix *N* × *A* (*P*) as Equation (1):(1)$$X=TP\prime +F=\overset{A}{\underset{a=1}{\sum }}t_{a}{p^{\prime }}_{a}+F$$
where *X* is the spectral data matrix, *T* is the score matrix for *X*, *P* is the loading matrix for *X*, *F* is the residual or model error matrix, *t_a_* is the sample score vector on each PC for *X*, and *p_a_* is the variable loading vector on each PC for *X*. In this study, the share of the first principal component (PC1), the second principal component (PC2), the third principal component (PC3) and the fourth component were 55%, 18%, 8%, and 6%, respectively. In total, the cumulative share rate of these four components reached 87.00%. To avoid under-fitting of the prediction model due to lack of components, and to prevent over-fitting due to information of redundant components, finally 14 main components were selected as input to the prediction model of the amount of pesticide residues in tomatoes.

### 2.6. Prediction Models

#### 2.6.1. PLSR

PLSR is a method for relating two matrices X (predictor) and Y (response), by a linear multivariate model, which also models the structure of X and Y [46]. It works well for analyzing large, noisy, and collinear data. In this model, by increasing the number of variables and related observations, the accuracy of the model parameters improves [47]. This method, the least squares solution, is applied to several orthogonal components that are a linear combination of independent variables and are created alternately with the aim of maximizing the covariance of the linear transformation of independent variables and dependent variables. It is very important to select the main factor when using PLSR for regression analysis. Wrong selection of the number of main factors causes the model to under-fitting or over-fitting, thus reducing the model prediction accuracy [48]. In this study, the mentioned method in the wavelength range of 300–1100 nm was used for modeling and analysis of spectral data. The fully cross-validation method was used to enhance the selection and the number of main factors RF, SPA, and PCA were 28, 14, and 14, respectively.

#### 2.6.2. BP-ANN

BP-ANN, a multilayer feed-forward neural network trained by the post-propagation error algorithm, is today the most widely used reductive neural network [40,49]. In this paper, a BP feed-forward neural network with one and two hidden layers was modeled. “tansig”, “logsig”, and “purlin” were used in the hidden and output layers as transfer functions. The training function used in this model was “trainlm” and the maximum number of repetitions was 3000. The optimal number of hidden layer neurons for RF-BP, PCA-BP, and SPA-BP combined models was obtained by trial-and-error method, 8, 12, and 14, respectively.

### 2.7. Model Validation

Validation methods are important to assess calibration precision and avoid data over-fitting. The predictive power of a calibration model can be evaluated by the R, RMSEP and RMSEC between the predicted value and the measured value in the validation set [50]. In this research, we used R and RMSEC-RMSEP values to evaluate the accuracy and overall strength of the model, respectively. These indicators are defined as follows:(2)$$R=\sqrt{\frac{\sum_{i=1}^{n}{\left({\hat{y}}_{i}-y_{i}\right)}^{2}}{\sum_{i=1}^{n}{\left({\hat{y}}_{i}-y_{mean}\right)}^{2}}}$$
(3)$$RMSEC=\sqrt{\frac{1}{n_{c}}\overset{n_{c}}{\underset{i=1}{\sum }}{\left({\hat{y}}_{i}-y_{i}\right)}^{2}}$$
(4)$$RMSECV=RMSEP=\sqrt{\frac{1}{n_{p}}\overset{n_{p}}{\underset{i=1}{\sum }}{\left({\hat{y}}_{i}-y_{i}\right)}^{2}}$$

${\hat{y}}_{i}$: Predicted value of ith observation.

$y_{i}$: Measured value of ith observation.

$y_{mean}$: Mean of the prediction or calibration set.

$n,n_{c},n_{p}$: The number of observations in the data set, calibration and prediction set, respectively.

In general, a good model should have higher correlation coefficients, lower RMSEC, lower RMSEP [51,52].

## 3. Results and Discussion

### Pre-Processing Spectra

Due to the presence of noise in the initial and final parts of the diagram of absorption spectra of tomato samples with different concentrations of pesticides, the spectrum range from 460–1050 nm was considered (Figure 6). The following 10 spectral preprocessing methods were applied to stabilize the models: Moving average, gaussian filter, median filter, S-Golay, Maximum normalize, derivative-S-Golay, SNV, MSC, (Gaussian filter) + (median filter), Normalize + Gaussian.

The residual reference values of the pesticide obtained by the GC-MS destructive test are between 42.9–“n.d” percent. Moreover, according to the prediction results of the combined models, the values of pesticide residues in the calibration and prediction data were between “n.d” up to 62.75%. The spectral diagram for tomatoes with different concentrations of pesticides is shown in Figure 6. In the diagram, the peak points in the visible and infrared region are closely visible. The peak points between 650–700, 750–800, and 960–1000 are related to the absorption of red pigments, the second and third overtone vibrations of OH and the first and second overtone vibrations of OH are related to water absorption. The results of PLS models obtained with different preprocessing methods to predict the Profenofos pesticide residues in tomato samples were shown in Table 3. Most of the developed calibration models had an acceptable ability to predict pesticide residues in samples with an RCV above 0.8. However, the best prediction results were obtained using the PLS model based on the Smoothing + moving average method (Rcv = 0.92, RMSECV = 4.25). Hence, this model was selected for further analysis. Shan et al. (2020), Soltani et al. (2021), Yi et al. (2010) and Sharabiani et al. (2019) also used the method used in this study to predict the amount of soil atrazine uptake, residual pesticides in strawberries, the amount of nitrogen in orange leaves and the amount of wheat protein, respectively, and achieved acceptable results [16,18,53,54].

Figure 7 shows the correlation diagrams of the predicted values versus the main values of the models used.

The use of NIRS technology in the detection of pesticide residues in fruits and vegetables, as well as their qualitative prediction, provides the researcher with a myriad of spectral data for analysis. Large amounts of spectral data complicate analysis, prediction errors, as well as over-fitting and under-fitting correlation curves. As a result, we need to reduce the data dimension. In this paper, it was found that the combined models used to predict the amount of Profenofos pesticide residues in tomato based on RF, SPA, and PCA can achieve the same performance using only a few characteristic spectra, and in some cases achieve better performance than the mode based on all spectral data (Figure 7). Some variables selected using the frog (28 wavelength) and SPA (14 wavelengths) algorithms were similar and the rest of the characteristic wavelengths were close to each other. Table 4 shows the results of model evaluation indicators. According to the results obtained in ANN-based combined models, using SPA algorithm with values of Rc = 0.989, Rp = 0.982, RMSEC = 0.141 and RMSEP = 0.166 and using total spectral data with values of Rc = 0.86, Rp = 0.81, RMSEC = 0.521 and RMSEP = 0.561, respectively, had the best and worst performance in predicting Profenofos pesticide in tomatoes. Also, in PLS-based combined models, the modes of using SPA, RF, PCA, and total spectral data had the best and worst performance in predicting, respectively. In general, according to the results obtained in terms of validation parameters, the best model proposed in this paper is the SPA-ANN model.

In a similar study the feasibility of using NIRS to detect the soluble solids content (SSC) of Malus micro malus Makino were studied using SPA, SVR, PLSR, and BP-ANN. The comparison studies confirmed that the optimal fusion model of SPA-SVR had the best performance (R_C_ = 0.9629, R_P_ = 0.9029, RMSEC = 0.199, RMSEP = 0.271) [42]. In other study, a new method of variable interval selection based on random frog (RF), known as Interval Selection based on Random Frog (ISRF), is developed. The results show that the proposed method is very efficient to find the best interval variables and improve the model’s prediction performance and interpretation [55]. The results of various studies show that the use of soft computing has been an effective method in the qualitative diagnosis of products. This is evidence of the confirmation of the results obtained from our study.

## 4. Conclusions

In this paper, a rapid and non-destructive near-infrared method was used to predict the profenofos pesticide residues in tomatoes. Spectral data obtained with a spectroradiometer and reference data obtained by a gas chromatography equipment were used as input and output of the models used in this research, respectively. PCA, SPA, and RF algorithms were used to select the variable as input for artificial neural network (ANN) and PLSR. First, all spectral data without dimension reduction and then spectral data obtained from variable selection algorithms were used to predict the amount of pesticide in tomatoes. Afterwards, 8 combined modes (pls, ANN, PCA-ANN, RF-ANN, SPA-ANN, PLS-PCA, PLS-RF, and PLS-SPA) were developed for prediction. Finally, the prediction accuracy of different combined models was compared and the best case was introduced. Based on what was said in the previous sections of the paper, it was found that it is possible to predict the amount of pesticide residues in tomatoes using the spectrum in the range of 460–1050 nm. Accordingly, it was determined that the use of variable selection methods had a better performance in predicting the amount of pesticide residues than the use of all spectral data. Finally, according to the results of the validation parameters of the combined models used, the SPA-ANN combined model with values of Rc = 0.989, Rp = 0.982, RMSEC = 0.141 and RMSEP = 0.166 had the best performance in predicting Profenofos pesticide in tomatoes.

At the end of the article, it can be mentioned that NIRS technology, in addition to advantages such as a non-destructive method, low cost measurement, high speed and online uses in the processes of quality determination. However, this method has some disadvantages, which are: NIRS requires chemometric techniques to provide the reference data for calibration and validation of experimental results, and the large number of samples with large variations to extract accurate information. 

## Figures and Tables

**Figure 1 sensors-21-03032-f001:**
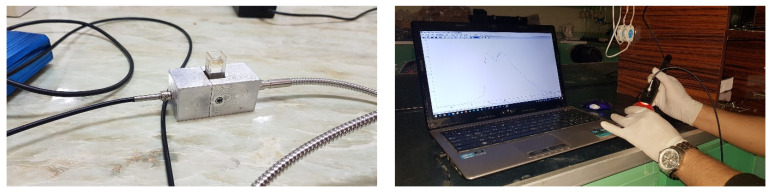
Measurement of Vis/NIR spectra of tomato samples in reflection mode and pesticide in passing mode.

**Figure 2 sensors-21-03032-f002:**
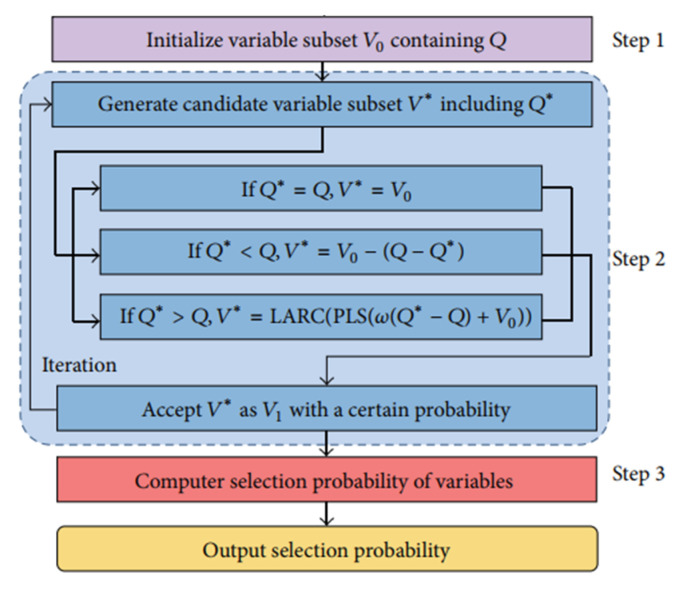
Flowchart of a random frog algorithm.

**Figure 3 sensors-21-03032-f003:**
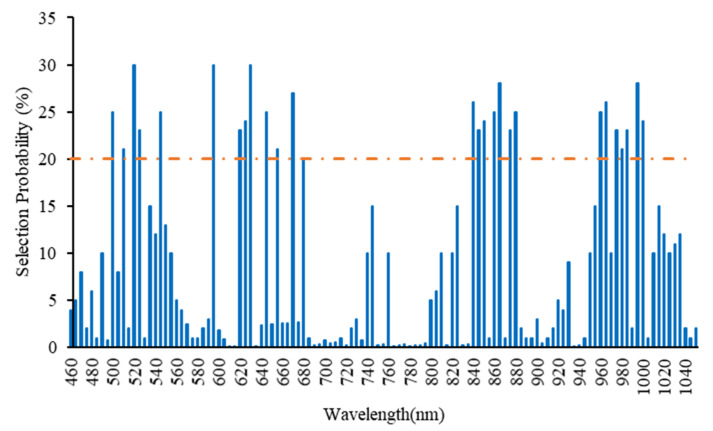
The result of extracting useful wavelengths using the RF algorithm.

**Figure 4 sensors-21-03032-f004:**
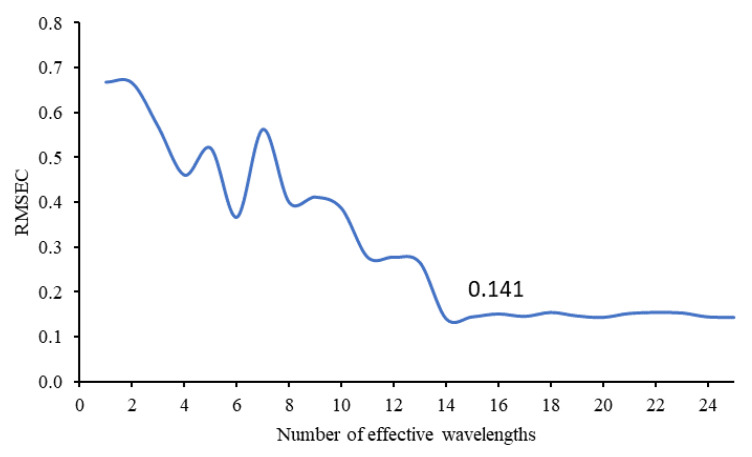
Change in RMSEC and Number of effective wavelengths.

**Figure 5 sensors-21-03032-f005:**
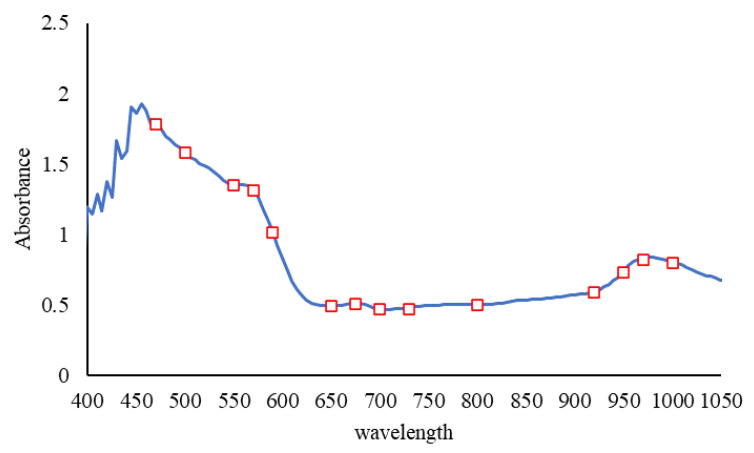
Selected bands by SPA.

**Figure 6 sensors-21-03032-f006:**
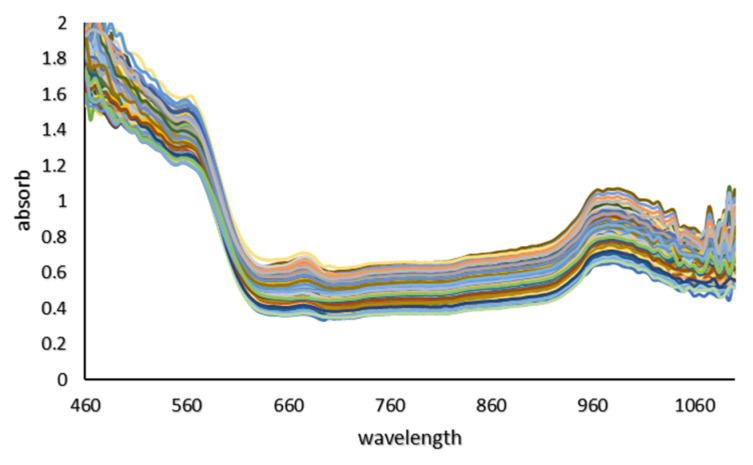
Absorption spectra of tomato samples with different concentrations of pesticides.

**Figure 7 sensors-21-03032-f007:**
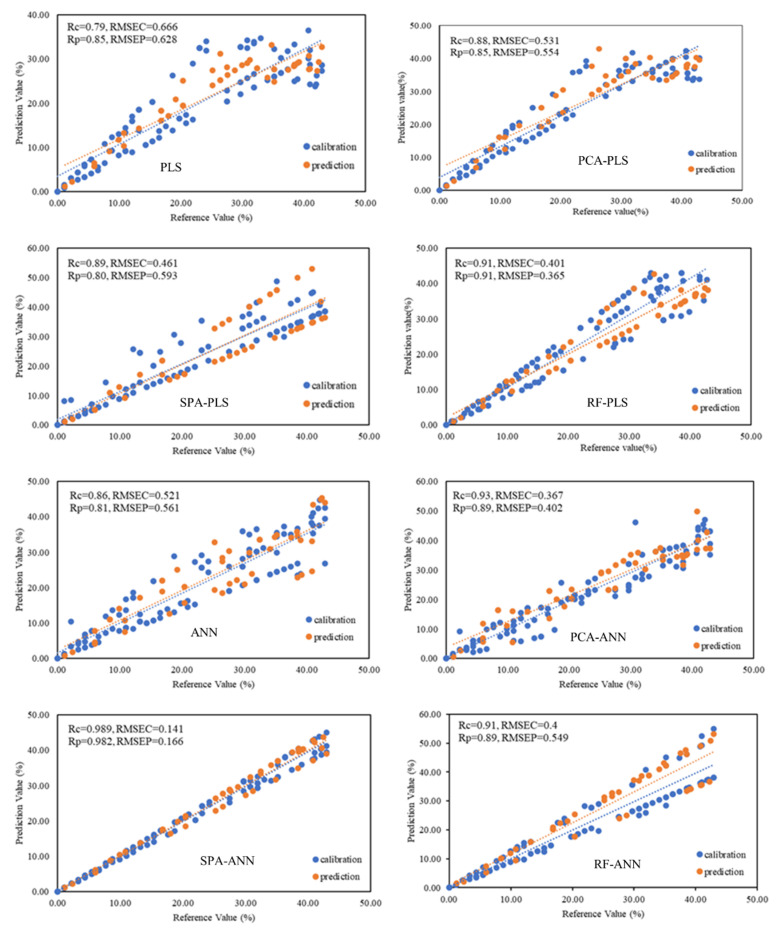
Correlation diagrams of the predicted values versus the main values of the models used.

**Table 1 sensors-21-03032-t001:** GC run conditions.

Analytical Column	HP-5 ms Ultra Inert 30 m × 250 μm, 0.25 μm (p/n 19091S-433UI)
Injection volume	1 μL
Injection mode	Spitless
Inlet temperature	280 °C
Liner	UI, split less, single taper, glass wool (p/n 5190-2293)
Plated seal kit	Gold Seal, Ultra Inert, with washer (p/n 5190-6144)
Carrier gas	Helium, constant flow, 1 mL/min
Oven program	60 °C for 1 min
	then 40 °C/min to 170 °C
	then 10 °C/min to 310 °C
	then hold for 2 min
Transfer line temperature	280 °C

**Table 2 sensors-21-03032-t002:** Reference values (mean, standard deviation (SD) and range) for profenofos content (mg/kg).

		Profenofos (mg/kg)		
	Number	Range	Mean	Standard Deviation
calibration	112	n.d. *–42.9	14.0	10.1
validation	48	n.d.–34.0	13.7	8.9

* Not detected.

**Table 3 sensors-21-03032-t003:** Results of different preprocessing methods for predicting Profenofos residues.

Pre-Processing	RMSE_CV_	R_CV_	LV
No preprocessing	5.7129	0.8609	15
Smoothing-moving average	4.2562	0.9254	13
Smoothing-gaussian filter	4.2680	0.9251	14
Smoothing-median filter	5.2481	0.8847	13
Smoothing	4.1379	0.9295	15
Maximum normalize	5.5788	0.8679	11
1derivative (S-Golay)	7.6328	0.7522	15
SNV	6.8656	0.7978	13
MSC	7.1441	0.7828	15
(Smoothing-Gaussian) + (smoothing median)	7.0276	0.7778	11
Normalize + Gaussian	5.9218	0.8490	10

**Table 4 sensors-21-03032-t004:** Results of validation parameters of combined models.

Combined Models	Validation Parameters			
	Rc	Rp	RMSEC	RMSEP
PLS	0.79	0.85	0.66	0.62
PCA-PLS	0.88	0.85	0.53	0.55
SPA-PLS	0.89	0.80	0.46	0.59
RF-PLS	0.91	0.91	0.40	0.36
ANN	0.86	0.81	0.52	0.56
PCA-ANN	0.93	0.89	0.36	0.40
SPA-ANN	0.98	0.98	0.14	0.16
RF-ANN	0.91	0.89	0.40	0.54

## Data Availability

Data is contained within the article.

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
