# Peer review of "Evaluation of Different Models for Non-Destructive Detection of Tomato Pesticide Residues Based on Near-Infrared Spectroscopy"

_sensors, 2021, doi:10.3390/s21093032_

Round 1

Reviewer 1 Report

The article is interesting and well-written. It's logically arranged and carefully drafted. The results of the research have been correctly described. Formulated conclusions are correct.

However, I have one remark. Namely, the authors should write about the advantages and disadvantages of the detection of tomato pesticide residues method used.

Reviewer 2 Report

Dear authors in the current research paper  a rapid and non-destructive near-infrared method was used to predict 3 the profenofos pesticide residues in tomatoes. The samples 180 were collected from one greenhouse condition. The experiment is interesting and well designed however the major drawback of this study that the calibration and especially the validation dataset belong to a single experiment which does not allow strong conclusions. I urge the authors to run additional experiments under different growing seasons. For instance, it is not mentioned whether the 180 samples came from a spring-summer tomato growing season or summer-autumn growing seasons. the effects of the growing seasons or other agricultural practices may affect strongly on the validation data set. 

Reviewer 3 Report

First of all, I congratulate the authors for the interesting work entitled "Evaluation of different models for non-destructive detection of tomato pesticide residues based on near-infrared spectroscopy". I consider that the work is relevant in practice, and that it is well presented. Now I have some suggestions to improve the manuscript:

  • Separate the Introduction correctly into paragraphs. The continuous way in which it is makes the reading cumbersome and not very fluent.
  • Improve grammar and spelling.
  • Table 2: significant figures are not to be used correctly.
  • In a whole paragraph of the Results and Discussion section, the authors make more a methodological description than the observations or a contribution to the interpretation or understanding of them (between lines 307-316): "In the study, outlier data were first removed using the Monte Carlo cross-validation method. The obtained 120 data were divided into two categories: calibration (70%) and prediction (30%). Ten spectral preprocessing methods were applied to stabilize the models and smoothing + moving average preprocessing method was selected as the best method. After analyzing the original data information, the main components obtained by PCA and the characteristic wavelength obtained from RF and SPA were used as input variables for ANN and PLSR models. First, PLSR and ANN models were used for prediction using all the spectral data obtained. Next, each method of variable determination (RF, SPA and PCA) in combination with PLSR and ANN models was used to predict the amount of pesticide residues in tomatoes. Finally, 8 models were developed to predict the amount of Profenofos pesticide residues. "
  • The use of present when should use past the description of the results, for example between lines 317-318: "The residual reference values ​​of the pesticide obtained by the GC-MS destructive test are between 42.9 -" n. d "percent."
  • The discussion should be enriched with the contrast of relevant previous works.

Round 2

Reviewer 2 Report

Unfortunately, the authors were not able to raise my concerns about the robustness of the data set. Again the experimental design and the manuscript are good. However, additional experiments are required for validation. The authors claimed that additional experiments will be done in the future. For the above-mentioned reasons I ask the AE to make the final decision based on my concerns and the recommendations of the other reviewer/s